# Targeted proteomics using stable isotope labeled protein fragments enables precise and robust determination of total apolipoprotein(a) in human plasma

Andreas Hober [1,2], Mirela Rekanovic[3], Björn Forsström[1,2], Sara Hansson[3],
David Kotol[1,2], Andrew J. Percy[4], Mathias Uhlén[1,2], Jan Oscarsson[5], Fredrik Edfors[1,2○],
Tasso Miliotis [3○] *

1 Science for Life Laboratory, Solna, Sweden, 2 Division of Systems Biology, Department of Protein
Science, School of Chemistry, Biotechnology and Health, The Royal Institute of Technology (KTH),
Stockholm, Sweden, 3 Translational Science and Experimental Medicine, Cardiovascular, Renal and
Metabolism, IMED Biotech Unit, AstraZeneca, Gothenburg, Sweden, 4 Department of Applications
Development, Cambridge Isotope Laboratories, Inc., Tewksbury, Massachusetts, United States of America,
5 Late-stage Development, Cardiovascular, Renal and Metabolism, Biopharmaceuticals R&D, AstraZeneca,
Gothenburg, Sweden

○ These authors contributed equally to this work.
* tasso.miliotis@astrazeneca.com

org/10.1371/journal.pone.0281772

Laboratory, UNITED STATES

**Data Availability Statement:** All raw data files and
extracted chromatograms can be accessed at
Panorama: https://panoramaweb.org/apoa.url,

## Abstract

Lipoprotein(a), also known as Lp(a), is an LDL-like particle composed of apolipoprotein(a)
(apo(a)) bound covalently to apolipoprotein B100. Plasma concentrations of Lp(a) are highly
heritable and vary widely between individuals. Elevated plasma concentration of Lp(a) is
considered as an independent, causal risk factor of cardiovascular disease (CVD). Targeted
mass spectrometry (LC-SRM/MS) combined with stable isotope-labeled recombinant pro-
teins provides robust and precise quantification of proteins in the blood, making LC-SRM/
MS assays appealing for monitoring plasma proteins for clinical implications. This study
presents a novel quantitative approach, based on proteotypic peptides, to determine the
absolute concentration of apo(a) from two microliters of plasma and qualified according to
guideline requirements for targeted proteomics assays. After optimization, assay parame-
ters such as linearity, lower limits of quantification (LLOQ), intra-assay variability (CV: 4.7%)
and inter-assay repeatability (CV: 7.8%) were determined and the LC-SRM/MS results were
benchmarked against a commercially available immunoassay. In summary, the measure-
ments of an apo(a) single copy specific peptide and a kringle 4 specific peptide allow for the
determination of molar concentration and relative size of apo(a) in individuals.

## Introduction

Lipoprotein(a) (Lp(a)) comprise of a low-density lipoprotein (LDL)-like particle with apolipo-
protein (a) (apo(a)) covalently attached to apolipoprotein B100 by a single disulfide bond. Apo
(a) is structurally different from any other apolipoproteins and contains a hydrophilic and

username: panorama+reviewer42@proteinms.net, password: xWuIHYpN Additionally, all raw files have been uploaded to ProteomeXchange with the data set id PXD026976.

**Funding:** The author(s) received no specific funding for this work.

**Competing interests:** The authors of this manuscript have the following competing interests: M.U. is a co-founder of Atlas Antibodies AB. This does not alter our adherence to PLOS ONE policies on sharing data and materials.

carbohydrate-rich structure with no amphipathic helices [1, 2]. Apo(a) is a homolog of plasminogen and both genes code sequences for loop structures stabilized by intrachain disulfide bonds, so-called kringle (K) domains. Within the apo(a) gene two different kringle domains are present, namely K4 and K5. The apo(a) gene has ten different types of plasminogen-like K4 domains, referred to as K4 type 1 through 10. All K4 domains except for K4 type 2 ($K4_2$) are present as single copies. In contrast, $K4_2$ repeats are present as multiple copies that vary from 3 to >40 copies [3–7]. Hence, apo(a) is highly polymorphic in size because of the number of $K4_2$ repeats, resulting in >40 isoforms with molecular masses ranging between 200 kDa and 900 kDa.

Due to the extensive LPA allele size variability, plasma Lp(a) in heterozygous individuals most often carry two differently sized apo(a) isoforms [8]. An individual may carry a small and a large isoform, in any combination thereof, and approximately 20% of all subjects express only one isoform of the protein [9]. The circulating levels of Lp(a) are primarily determined by the association between $K4_2$ repeats and Lp(a) levels, but a variation in Lp(a) levels has been observed for isoforms of the same size [10–14].

High Lp(a) levels is a risk factor for coronary artery disease (CAD), myocardial infarction (MI), stroke, peripheral arterial disease (PAD), calcific aortic valve disease (CAVD), and heart failure (HF) [10, 11, 15–25]. Mendelian randomization studies have shown that low numbers of $K4_2$ repeats and high Lp(a) levels are associated with CVD [10]. These associations have catalyzed research and development efforts to introduce therapies to reduce the plasma levels of Lp(a) [26–28].

The size heterogeneity of Lp(a) has been the most challenging hurdle to overcome for the development of immunoassays for the accurate measurement of Lp(a) in plasma. Marcovina *et al.* [29] have reviewed the challenges involved in selecting specific antibodies directed to apo(a) due to the variability in size of apo(a) and to develop an assay calibrator that has structural characteristics that resemble the analyte. Selecting an appropriate calibrator is practically impossible due to the intra- and interindividual high degree of size variation in apo(a). Furthermore, because of this heterogeneity, Marcovina *et al.* [29] suggested that quantitative Lp(a) levels from assays should be reported in apo(a) particle number as nmol/l and not mg/dl, provided the variability in apo(a) size. Due to the need for precise values represented in standardized and comparable units, several targeted LC-MS/MS approaches have been developed for the quantification of apolipoprotein(a) by targeting selected proteotypic peptides. These have been constructed relying solely on spike-in of peptide standards, which enable the peptide concentration to be determined precisely by isotope dilution. These assays have been rigorously assessed in terms of accuracy and precision and designed to circumvent the issues of the repetitive sequence of apo(a) [30–32]. A common obstacle when working with peptide standards is their dependence on proteolysis yields to deliver robust, accurate and precise quantification. This has been highlighted both by Marcovina *et al.* and Blanchard *et al.*, in the context of accurate quantification. To address these inherent targeted proteomics issues, rigorous evaluation of digestion kinetics or the use of commercial digestion kits have been used, but this puts a new constraint on the assay in terms of confinement to specific digestion protocols [32, 33]. In our previous study [34], a novel type of internal standard called Stable Isotope Standard (SIS) Protein Epitope Signature Tags (SIS PrESTs), which are derived from the Human Protein Atlas project, was described in the context of analysis of clinical samples [35, 36]. These standards are based on recombinant protein fragments that release their labeled peptides upon digestion and can account for very short digestion protocols with preserved quantitative precision [35, 37]. The SIS PrESTs were used to quantify a multiplexed panel of 13 apolipoproteins, including apo(a), in human plasma using Liquid-Chromatography coupled with Selective Reaction Monitoring Mass Spectrometry (LC-SRM/MS). There are significant benefits to

using targeted mass spectrometry in conjunction with protein-level rather than peptide-level spike-inof standards and SIS PrESTs are an excellent substitute for full-length protein standards in terms of precision and accuracy [34, 38]. However, the SIS PrEST design for apo(a) that was used in our previous study quantified the amount of K4 repeats rather than the apo(a) concentration in human plasma since the target peptide sequence is found in the $K4_2$ domain. In this study, we describe the use of a new SIS PrEST apo(a) standard that circumvents this issue by selecting peptides belonging to the non-repeated region of apo(a). The proteotypic peptide is situated in the protease domain of apo(a) and thereby enables measurement of the molar apo(a) levels in plasma. The use of LC-SRM/MS in combination with an internal standard independent of apo(a) isoforms eliminates the dependence on antigen-antibody reactions, calibrators and standards related to immuno-assays for quantification of apo(a). By also quantifying the peptides belonging to the K4 domains, the measurements can be used to obtain the mean number of K4 domains that are present in the apo(a) molecules in the sample and thereby estimate the average size of apo(a) [39].

LC-SRM/MS assays have emerged as a viable alternative to immunoassays offering high specificity, multiplexity and excellent precision with automated data acquisition [40]. Herein, we describe the development of an LC-SRM/MS method that has been optimized and evaluated with standard procedures for analytical method validations. This includes evaluation of linearity, limit of quantification (LOQ) and repeatability, which is in accordance with a fit-for-purpose approach for targeted mass spectrometry-based assay development [41]. A sample volume of only 2 μl human plasma was sufficient for accurate and precise quantification of apo(a). The protocol involves tryptic digestion and the generated peptides act as surrogate markers for the protein(s) of interest, which are subsequently analyzed by LC-SRM/MS [42, 43]. The targeted mass spectrometric assay was compared to a commercially available solid-phase two-site enzyme immunoassay based on the sandwich technology, in which two monoclonal antibodies, directed against separate antigenic determinants on the apo(a) molecule, were used. In a recent study using a well-characterized healthy sample cohort (The Swedish Science for Life Laboratory SCAPIS Wellness Profiling (S3WP)) of 101 healthy individuals, it was shown that the adult blood levels of many proteins are determined at birth by genetics [44]. Plasma from the S3WP cohort was used to determine the apo(a) levels and how the number of K4 domains influences plasma levels of apo(a).

## Materials and methods

### SIS PrEST standards

The internal Stable Isotope Standard Protein Epitope Signature Tag (SIS PrEST) was designed to contain proteotypic peptides of the protease domain of apo(a) (S1 Table) [45]. Atlas Antibodies AB (Stockholm, Sweden) performed recombinant protein production, purification, and standard quantification. The stock concentration of the SIS PrEST was 23.2 μM. The SIS PrEST was diluted to a concentration of 20 nM in LC-MS grade water. LC-SRM/MS assays were developed with the following constraints: Peptide ions with a length of 5–25 amino acids and a precursor charge state z = 2,3 and product ion charge state z = 1,2 were screened. Only b- and y-ions were included in the analysis.

### Biological samples

Plasma ($K_2$-EDTA) for method development was collected from five healthy non-obese Caucasian volunteers, three female and two male donors, obtained from the AstraZeneca R&D Gothenburg biobank. Moreover, an additional ten randomly chosen human plasma samples also obtained from the AstraZeneca R&D Gothenburg biobank of healthy volunteers were

used for the immunoassay measurements. Informed consent was obtained from all subjects. The study was performed according to local ethical regulations following approval from the regional ethics committee "Regionala etikprövningsnämnden i Göteborg" in Gothenburg, Sweden. The plasma samples were pooled and aliquoted into 0.5 ml Protein LoBind Microcentrifuge Eppendorf tubes that were subsequently stored at -80˚C and also used as quality control samples. A well-characterized healthy sample cohort (The Swedish Science for Life Laboratory SCAPIS Wellness Profiling (S3WP) was analyzed [46]. Informed consent was obtained for all participants. The study was performed in accordance with the declaration of Helsinki and the study protocol was approved by the Ethical Review Board of Göteborg, Sweden (Regionala etikprövnignsnämnden, Gothenburg, Dnr 407–15, 2015-06-25). All samples were de-identified and randomized prior to the proteomics analysis.

### In-solution digestion protocol

Samples and SIS PrEST standards were denatured and reduced in a final concentration of 6.7 mM TCEP, 9 M Urea, 300 mM Trizma buffer (pH 8.0) at 37˚C (650 rpm) for 1 hour in a ThermoMixer (Eppendorf, Hamburg, Germany). Subsequently, alkylation was performed by the addition of 2-chloroacetamide (CAA) to a final concentration of 20 mM followed by incubation in the dark for another 30 min at room temperature. The samples were diluted by addition of 100 mM Tris buffer (pH 8.0) to reach a urea concentration of 0.7 M before addition of trypsin with a substrate to enzyme ratio of 30:1. The tryptic digestion was performed overnight for 17 h at 37˚C (650 rpm) in a ThermoMixer. The digestion was terminated by quenching with formic acid (FA) with a final concentration of 1%.

### Automated solid-phase extraction

Solid-phase extraction (SPE) of the digested plasma samples was performed on a Bravo Agilent AssayMAP liquid handler robotic system using reversed-phase (RP-S) 5 µl cartridges (G5496-60033, Agilent technologies, USA) in order to desalt and concentrate the samples. The method has been described in detail elsewhere [34]. Briefly, a total amount of approximately 100 µg of peptides were loaded on the RP-S cartridges. After the run, the peptides were eluted (10 µl of 60% acetonitrile, 0.1% TFA) and collected into an elution plate (Eppendorf PCR plate, Cat no. 0030129300) where the wells contained 90 µl of 0.1% FA, resulting in a total peptide concentration of about 1 µg/µl.

### Standard curves

The SIS PrEST was serially diluted (n = 12) into the plasma pool (five healthy volunteers) in a 2-fold manner, covering a range from 1 µM to 0.49 nM. Each calibration point was digested in triplicates, as outlined above, and analyzed using the developed LC-SRM/MS method. Standard curves were established by using the median ratio calculated between the standard peptides and the endogenous peptides for each spike in level. Standard curves were established according to CPTAC guidelines [47].

### Stability study

Three freeze/thaw cycles were performed with aliquots of pooled human plasma stored at -80˚C for the analyte stability study. Three aliquots were thawed in a water bath at room temperature. One aliquot of thawed plasma was put in the refrigerator awaiting digestion. The remaining aliquots of thawed plasma were frozen on dry ice and stored at -80˚C for one hour. The procedure was then repeated for the second and third aliquot at different days. Thereafter,

each thawed plasma aliquot was digested in triplicate as described above, subsequently followed by SPE and LC-SRM/MS analysis.

## Assay evaluation in the repeatability experiment

The single-laboratory precision of the apo(a) assay was calculated as the intraday and interday variation between samples, respectively, and were determined as the coefficient of variation in percentage (CV%). Triplicate parallel digestions were performed and subjected to LC-SRM/MS analysis across five consecutive days, where three technical replicates were measured for each sample on each day resulting in totally 45 LC-SRM/MS runs.

For estimation of the intraday CV the median (m) and standard deviation (sd) of three injections per sample were calculated as described by Hober *et al.* [34]. The acceptance criteria consisted of a CV value of $\leq 10\%$.

## Immunoassay

A commercially available apo(a) enzyme linked immunosorbent assay (ELISA) (Cat. No. 10-1106-01, Mercodia AB, Uppsala, Sweden) was used for the quantification of apo(a) in human plasma samples. The immunoassay is a sandwich ELISA with one capturing apo(a) antibody and one detecting apo(a) antibody. The measurements were conducted according to the manufacturer´s instructions of ten randomly selected human plasma samples from the AstraZeneca R&D Gothenburg biobank. The concentrations of apo(a) in the plasma samples were measured with the Spectramax M2 (Molecular devices) at 450 nm.

The kit measured the apo(a) concentration in units per liter (U/l) using five ready-to-use calibrators. A conversion factor was provided by the vendor to convert the concentration to mg/dl (1 U/l = 0.1254 mg/dl).

The ELISA was qualified by confirming dilution linearity, spike recovery and determination of the most suitable dilution for the samples by using two controls selected from the plasma samples. Thereafter, the confirmed dilution was used, and samples were analyzed and evaluated. Subsequently, the apo(a) levels determined by the ELISA assay were compared against the apo(a) levels determined by the SIS PrEST-based LC-SRM/MS assay.

## Determination of calibrator concentration

LC-SRM/MS was used to quantify the calibrators included in the ELISA kit (Mercodia AB) in order to be able to correlate the results of the immunoassay with the results of the developed apo(a) SIS PrEST-based LC-SRM/MS assay. According to the manufacturer, the concentration of the highest concentration calibrator (no. 4) was 5.14 U/l. Using a known amount of SIS PrEST this was transformed into fmol per μl plasma using the LC-SRM/MS assay. Calibrator no. 4 (5.14 U/l) was dissolved in 250 μl of water and serially diluted three times (dilution scheme 1:1) and each sample was subjected to digestion, SPE and LC-SRM/MS analysis. The mean value from the three samples obtained from the LC-SRM/MS analysis in nM was used to calculate the absolute calibrator concentration.

## LC-SRM/MS analysis of reference samples

Approximately 10 μg of peptides from SPE eluate were injected onto the LC-SRM/MS system. The LC system (Agilent 1290, Waldbronn, Germany) was interfaced to a triple quadrupole mass spectrometer (Agilent 6490, Santa Clara, CA, USA) using the Agilent Jet Stream flow ESI source operated at a positive ion mode. The peptide separation was performed on a Zorbax Eclipse XDB-C18 2.1 x 150 mm column packed with 1.8 μm silica particles with a pore size of

80 Å (Agilent Technologies, Santa Clara, CA). The column compartment was maintained at a temperature of 50˚C and the samples were refrigerated at 8˚C in the autosampler. The flow rate was kept at 400 μl/min and the separation was accomplished using gradient elution, where the used gradient conditions are presented in S2 Table. Mobile phase A consisted of 0.1% FA in water and mobile phase B consisted of 0.1% FA in acetonitrile (ACN). Followed by the peptide samples, a blank sample consisting of 0.1% FA was injected into the LC-SRM/MS system to monitor carryover effects.

## Digestion of samples from healthy individuals

Blood plasma from healthy individuals were collected in 6 mL EDTA tubes (Vacuette, 456243) and centrifuged immediately at 3000 rpm at room temperature. Following centrifugation, the plasma was transferred to 0.5 mL tubes (Sarstedt, 72.730.003) and frozen within 20 minutes. The samples were stored at -80˚C before being transferred to SciLifeLab for examination. A pool of SIS PrESTs was made and spiked to empty wells of a 96 well LoBind plate (0030129512, Eppendorf, Hamburg, Germany) according to Table 1. The plate was vacuum centrifuged to dry off any liquid. Ten times diluted plasma corresponding to two microliters of raw plasma was added to the plate of dried standards and the samples were digested following the same procedure as described. Each digest was then subjected to SPE using StageTips as described earlier [48].

## LC-SRM/MS analysis of healthy individuals

Approximately 10 μg of peptide amount was loaded onto an Ultimate 3000 (Thermo Fischer Scientific, Waltham, MA, USA) LC-system fitted with a 15 cm EasySpray analytical column (PN ES802A rev.2, particle size: 2 μm, pore size: 100Å, 150 μm x 15 cm, Thermo Fischer Scientific) and an Acclaim PepMap 100 trap column (PN 160454, particle size: 5 μm, pore size: 100 Å, 0.3 mm x 5 mm, Thermo Fischer Scientific). The peptides were eluted across a linear gradient using a 35 min method, with a flowrate of 3 μl/min and a mobile phase consisting of solvent A (3% ACN, 0.1% FA) and solvent B (95% ACN, 0.1% FA) and a gradient as described in S3 Table. The LC was coupled to a TSQ Altis (Thermo Fischer Scientific) operating with a cycle time of 0.5 seconds monitoring the transitions specified in S4 Table.

**Table 1. Composition of standard pool used for quantification of apolipoproteins in sample cohort.**

| HPRR ID | Gene | Amount per sample [pmol] |
|---|---|---|
| HPRR2190035 | LPA | 0.25 |
| HPRR260124 | APOA4 | 3.2 |
| HPRR2760373 | APOD | 3.3 |
| HPRR3340379 | APOM | 0.20 |
| HPRR3450266 | APOA1 | 37 |
| HPRR350023 | APOF | 0.018 |
| HPRR350088 | APOL1 | 0.12 |
| HPRR3720311 | APOB | 0.33 |
| HPRR3730489 | APOC1 | 1.5 |
| HPRR4130067 | APOC4 | 0.027 |
| HPRR4200068 | APOE | 0.63 |
| HPRR4320626 | CLU | 0.30 |
| HPRR4430020 | APOA2 | 5.2 |
| HPRR5000605 | LPA | 0.36 |

## Data analysis

All MS raw files were analyzed using Skyline (version 21.0.9.118) [49], in which the peak shape as well as the presence of both standard and endogenous peptides were assessed. This was done by calculating a rdtop-value for each peptide individually. Peptides not fulfilling these criteria (rdotp below 0.75) were excluded from the analysis. This provided a final data-set with a median rdotp > 0.99 The ratio between the spiked in standard and the endogenous peptides was used to calculate the molar concentration based on the known spike-in amount.

## Data availability

All raw data files and extracted chromatograms can be accessed at Panorama Public [50]: https://panoramaweb.org/apoa.url. Additionally, all raw files can be accessed through ProteomeXchange (PXD026976) [51].

## Results

### Assay generation

Two separate apo(a) regions were targeted by using two separate SIS recombinant proteins as illustrated in Fig 1A. The total apo(a) concentration was determined by one standard designed to cover the plasminogen domain (PD, purple), which is unique in the protein sequence. The second standard was directed towards the K4 domain (blue) to determine the total concentration of repeated K4 domains (Fig 1B). The total number of K4 domains in relation to the PD domain was determined by normalizing the concentration values using the SIS standards. The protein concentration determination was performed using the same principle as previously described by Hober $et\ al.$ [34]. The specific number of repeated $K4_2$-domains can be obtained by subtracting the number of peptides mapping to other K4 main types.

A set of seven theoretical proteotypic peptides from the PD was initially selected for this study (S5 Table). Two peptides were considered at risk of exhibiting post-translational modifications (according to Uniprot) and were discarded from the analysis, while the remaining five peptides were considered for further evaluation. This standard protein was produced, its peptides were screened, and assay parameters were optimized as previously described. (34, 35). The standards were spiked and serially diluted into pooled human plasma (n = 12) and analyzed by LC-SRM/MS. The peptides were validated over a wide concentration range by reverse standard curves (S1 Data) using the pooled human plasma as background. The results show that apo(a) is detectable across a broad dynamic range (Table 2), down to an estimated concentration of 3.5 nM using the peptide EAQLLVIENEVCNHYK$^{(+3)}$ for quantification (S1 Data). Here, the difference between the reported quantification and the known analyte dilution is known as the prediction error. The linearity of the peptide was thoroughly evaluated and the LOD, LOQ and assay linearity was assessed according to guideline requirements presented by The Clinical Proteomic Tumor Analysis Consortium (CPTAC) for targeted proteomics assays [41]. Calculations of the characteristics are based on a reverse curve approach [52]. The linear dynamic range for the target protein apo(a) was established by serial dilution of the SIS PrEST standard spiked into the pooled human plasma matrix at different concentrations, ranging from 0.49 nM to 1.0 μM. The standard curve and linearity measurements resulted in a linear response for apo(a) from 0.5 pmol/μl to 7.8 fmol/μl. Reverse standard curves were used to analyze all selected peptides from the SIS PrEST. Because of its sensitivity, determined by the lowest LOQ, the peptide EAQLLVIENEVCNHYK$^{(+3)}$ was chosen for the measurement of the plasminogen domain.

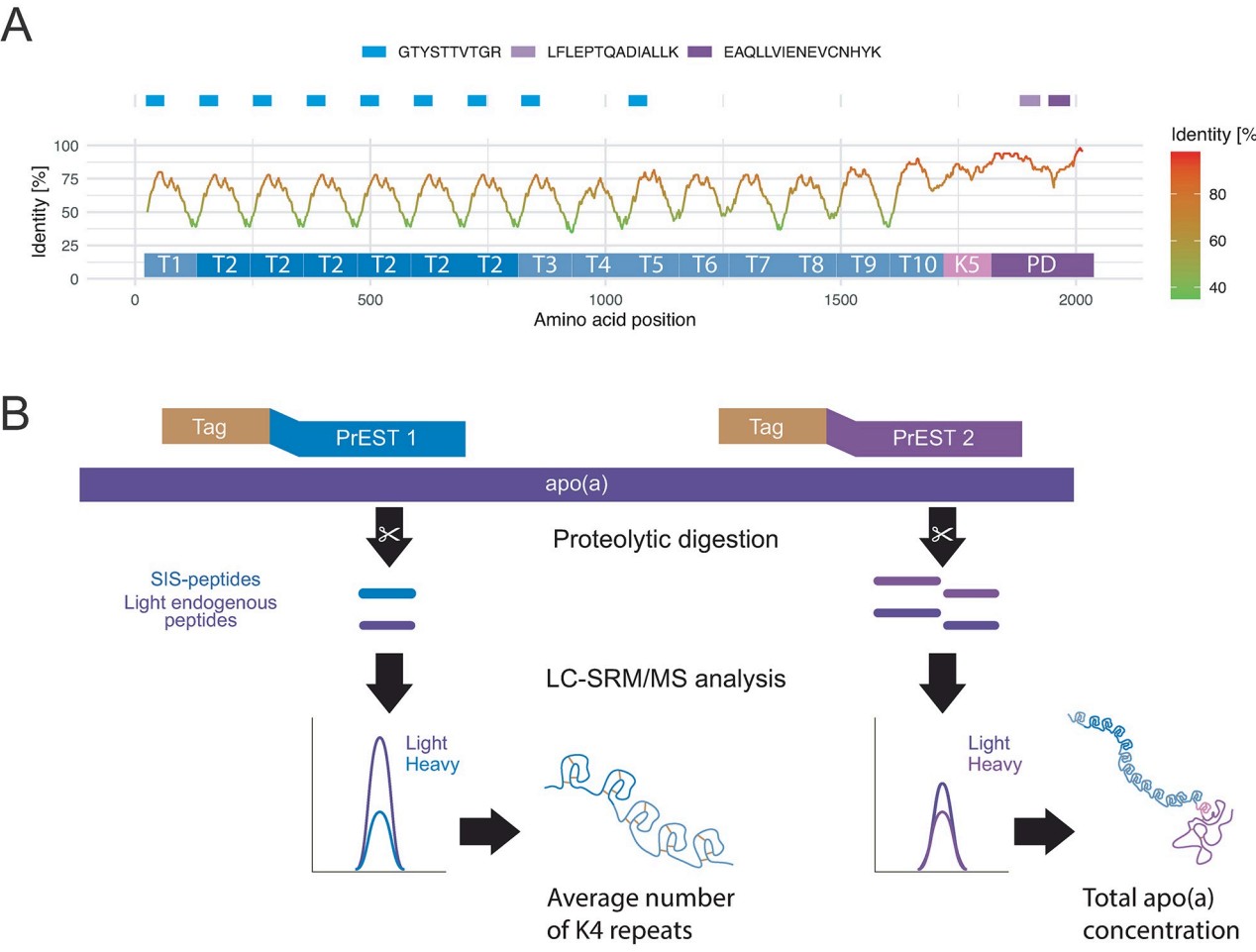

**Fig 1. A)** Visualization of the homology of apo(a) to other human proteins plotted as identity based on HsID50 and the transcript ENSP00000321334 (red = high, green = low). T1-T10 is used to denote the K4 domains and PD designates the plasminogen domain. Three different SIS PrEST peptides are mapped to the sequence at the top. **B)** A schematic visualization of how the SIS PrESTs are used to determine the number of K4 domains following quantification of total apo(a) and the K4 domain upon proteolytic digestion. One SIS PrEST is used to determine the molar concentration of peptides mapping to the K4 domains and one SIS PrEST is used to determine the molar concentration of apo(a). The ratio between the concentration from the K4 domain and the molar concentration of apo(a) corresponds to the number of K4 repeats.

## Assay repeatability

The apo(a) assay repeatability was investigated to determine the technical variation during sample preparation taking place at different days. Samples from one human plasma pool of healthy volunteers (n = 10) were thawed daily and prepared in triplicates over a total period of five days (Fig 2A). Samples were injected in three technical replicates and the sample median was used as the daily measurement. The assay's intraday variation was calculated as the mean of CVs obtained from triplicate samples analyzed within a single-day run. The interday variation was calculated from the triplicate measurements made across five days. The assay showed high repeatability (Fig 2A) with an intraday CV of 4.7% and an interday CV of 7.8%.

The robustness of the assay was investigated after repeated freeze-thaw cycles. Three samples from a pool of human plasma were subjected to three repeated freeze-thaw cycles over the course of 3 days. Samples were digested by trypsin in triplicates and the CV was calculated as the mean value of the median of three technical injections. The results show that apo(a) is stable after three repeated freeze-thaw events with a CV below 5% for each peptide (Fig 2B). This

**Table 2. Quantitative performance of EAQLLVIENEVCNHYK across a range between 490 fM to 1,000 nM in human plasma.**

| Conc. [nM] | Ratio to Standard | Standard deviation | CV [%] | Predicted conc. [nM] | Prediction error [%] |
|---|---|---|---|---|---|
| 1000 | 0.0169 | 0.00030 | 1.8 | NA | NA |
| 500 | 0.0485 | 0.0015 | 3.1 | 624. | 24.8 |
| 250 | 0.145 | 0.0040 | 2.7 | 225 | 10.0 |
| 120 | 0.335 | 0.0060 | 1.8 | 109 | 13.0 |
| 62 | 0.692 | 0.019 | 2.7 | 56.4 | 9.8 |
| 31 | 1.45 | 0.10 | 7.2 | 28.8 | 7.8 |
| 16 | 2.73 | 0.31 | 11.3 | 16.3 | 4.4 |
| 7.8 | 5.13 | 0.91 | 17.6 | 9.20 | 17.8 |
| 3.9 | 7.40 | 1.5 | 21.3 | NA | NA |
| 2 | 10.5 | 1.9 | 18.5 | NA | NA |
| 0.98 | 14.4 | 5.8 | 40.0 | NA | NA |
| 0.49 | 19.2 | 1.9 | 9.7 | NA | NA |

finding indicates that the protein can be quantified in multiple consecutive experiments while being thawed and frozen several times without the assay losing its quantitative accuracy.

## Accuracy of immuno-assays quantifying apo(a)

A commercial CE-IVD certified ELISA assay was obtained to verify the performance of the established LC-SRM/MS assay. Briefly, the Mercodia Lp(a) ELISA 10-1106-01 assay has been benchmarked against three different lots and concentrations of Bio-Rad Liquicheck Lp(a) control, which in turn have been quantified by five different methods, namely Abbott ARCHI-TECT cSystems (Biokit), Beckman Coulter IMMAGE, Binding Site SPAplus, Roche/Hitachi cobas cSystems and Siemens BN Series Nephelometers. This serum aliquot is used for quality control and equivalence determination for clinical assays. The CE-IVD certified reference material consists of a human serum containing stabilized triglyceride components of human

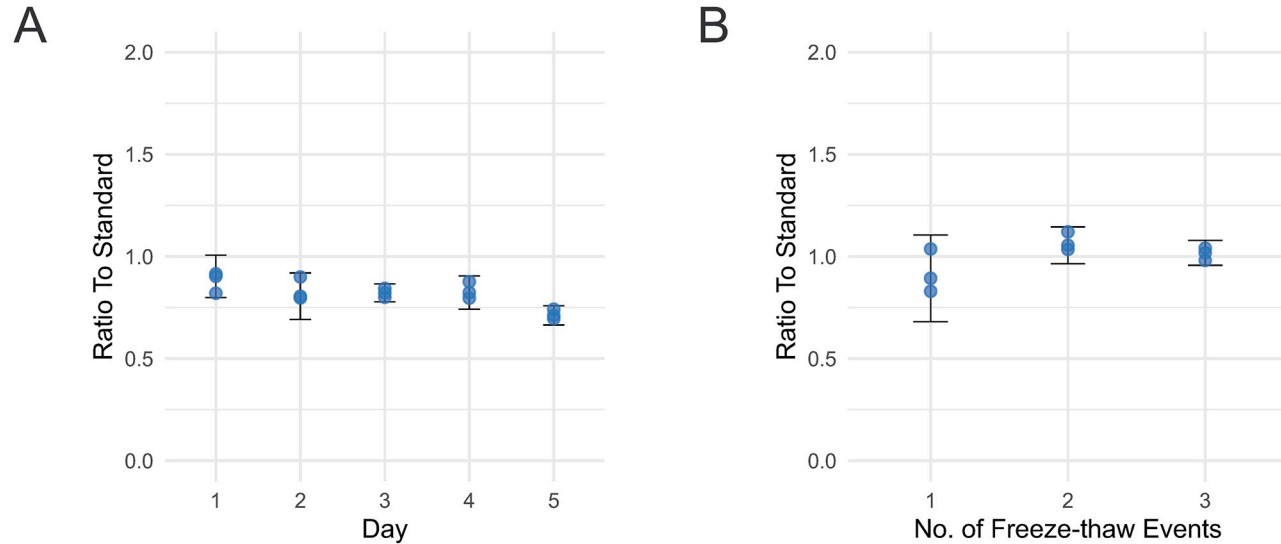

**Fig 2. A)** Repeatability data indicating the intraday variation during each day of the five-day period and the overall inter-day variation of the proteotypic peptide. **B)** Stability freeze-thaw data of three cycles for the proteotypic peptides. In both figures 2 times standard deviation interval is visualized with error bars.

blood plasma and stabilized apolipoproteins levels used as quality control material to monitor the ongoing precision of clinical laboratory systems. The results in U/L were converted to µg/µl using a specific conversion factor, determined by five different methodologies that are based on external controls (Bio-Rad Liquicheck Immunology Controls). The immunoassay is based on a number of calibrators that have to be selected for each sample cohort individually to quantify apo(a) with high precision and accuracy. One calibrator (no. 4) with a concentration of 0.129 µg/µl, was selected as the optimal range based on a pool of ten individuals. The established LC-SRM/MS assay towards apo(a) was used to quantify the calibrator and was determined to be 2.3 nM. This value was used to convert the concentrations obtained by ELISA to molar concentrations. The immunoassay was used to determine the concentration of apo(a) in 10 individuals (Table 3). The same samples were analyzed using the apo(a) LC-SRM/MS assay as described above. The immunoassay and the LC-SRM/MS method correlated with an $R^2$ of 0.92 (Fig 3). The immunoassay's accuracy was calculated based on the LC-SRM/MS result, and the antibody-based assay showed good accuracy in the concentration range that lies closer to the pooled plasma reference. However, a deviation between the methods was noted for the samples with plasma concentrations below LOQ of the LC-SRM/MS assay (Table 3).

## Profiling a wellness cohort of healthy individuals

A previously well-profiled cohort [46] of individuals without any known disease had previously been investigated using the SIS PrEST assay measuring the K4 repeats [48]. The same samples were used to quantify total apo(a) molar concentration and number of K4 domains. A subset of plasma samples (90 samples + 6 pools) was de-identified prior to quantification of apo(a). The final apo(a) assay was combined with a previously established assay capable of profiling 13 apolipoproteins in human plasma in multiplex [34], allowing for deeper apolipoprotein profile characterization. The final plate contained six replicates of pooled plasma (3 males, 2 females) used to assess the assay's technical variation.

The overall concentration of apo(a) ranged between 311 nM and 1.9 nM (Fig 4A). In order to determine the number of K4 domains within a sample, both the molar concentration of the peptide representing K4 domains (GTYSTTVTGR) and the molar concentration of the peptide EAQLLVIENEVCNHYK, corresponding to the total molar amount of circulating apo(a), were analyzed. By dividing the molar concentration of GTYSTTVTGR with the molar concentration of EAQLLVIENEVCNHYK, the average number of K4 repeats per apo(a) molecule in a sample was obtained. A constant of six repeats have to be added to the obtained number of K4 repeats to account for domains not possessing the GTYSTTVTGR peptide (Fig 1A). In the

**Table 3. Quantification results of apo(a) from ELISA and LC-SRM/MS.**

| Sample | ELISA [nM] | LC-SRM/MS [nM] | Difference [%] |
| --- | --- | --- | --- |
| Pool | 32.1 | 29.3 | 9.4 |
| S1 | 33.7 | 47.1 | -28.5 |
| S4 | 19.6 | 23.3 | -15.9 |
| S6 | 1.9 | 7.1 | -73.3 |
| S9 | 59.8 | 60 | -0.3 |
| S11 | 31.2 | 32.9 | -5.2 |
| S13 | 12.1 | 18.1 | -33.3 |
| S14 | 216.8 | 201.3 | 7.7 |
| S16 | 91 | 95.8 | -5 |
| S17 | 0.4 | 4.9 | -91.8 |
| S20 | 167.5 | 242.2 | -30.8 |

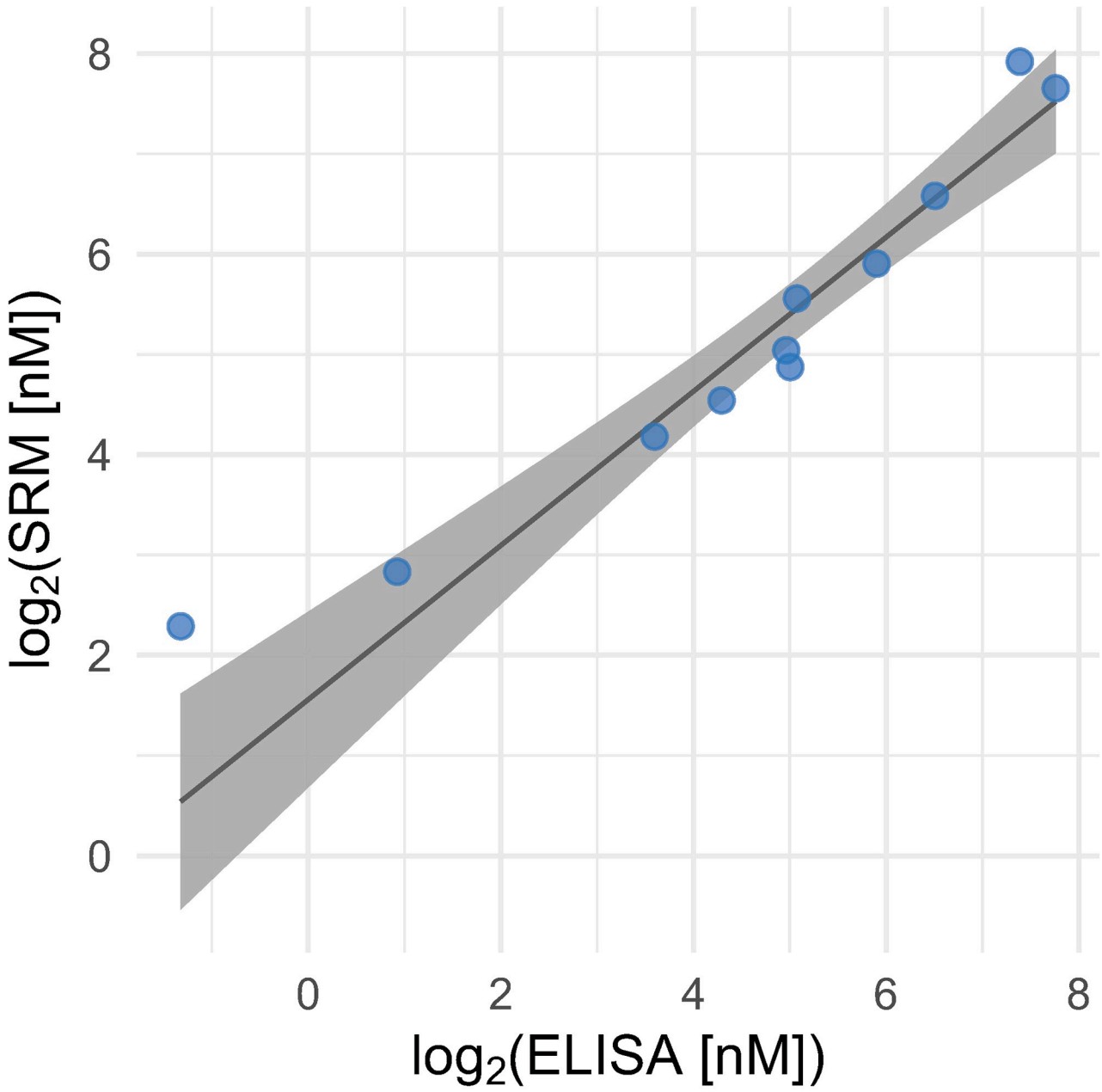

**Fig 3. Comparison between CE-IVD approved immuno-assay and the targeted proteomics method.** The ELISA result has been transformed to a molar concentration (from U/l) after quantifying calibrator no. 4 using LC-SRM/MS in triplicate measurements.

S3WP cohort, most individuals had between 14–21 K4 repeats (Fig 4B), which correspond to 5–12 $K4_2$ repeats. Fig 4C shows no evident correlation between the number of repeats and the molar concentration of apo(a). Interestingly, two different clusters appear in the analysis, which can be separated by a concentration cutoff of 47 nM. The cluster with values >47 nM shows a negative correlation (rho = -0.54, p = 0.013) previously described in larger cohorts [10] and can be defined by individuals with more than about 13–14 $K4_2$ repeats. The other cluster represents individuals with low $K4_2$ copy numbers, where no correlation between the number of $K4_2$ repeats and apo(a) concentration is observed.

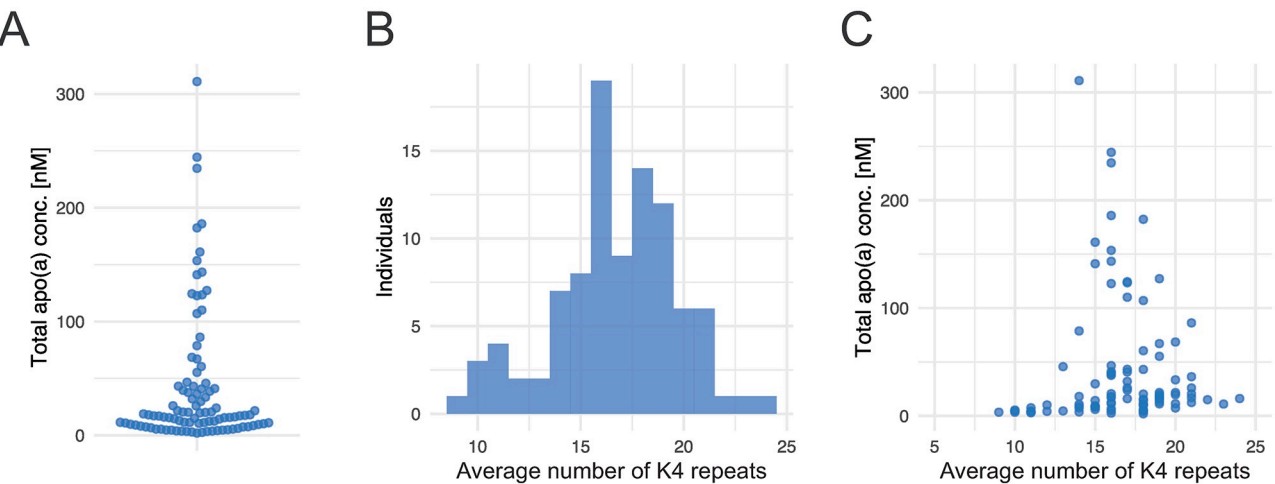

**Fig 4. Apolipoprotein plasma profiling of 90 individuals. A)** Beeswarm plot visualizing the total apo(a) levels among the 90 profiled subjects determined by the peptide EAQLLVIENEVCNHYK. **B)** Histogram of the average number of K4 domains present in the profiled samples, determined by the peptide GTYSTTVTGR. **C)** The average number of K4 repeats plotted against the total plasma concentration of apo(a).

## Discussion

The described method, using the specificity of targeted proteomics and mass spectrometry, allows the measurement of the molar concentrations of apo(a) as well as the average isoform size of the apo(a) protein in clinical studies. The assay shows a higher degree of repeatability and performance than a corresponding ELISA test that indirectly estimates the molar concentration of apo(a) in blood by using different references. The targeted mass spectrometry assay using spiked SIS PrESTs enables a direct measurement of peptides without any conversion factors and can be applied over large dynamic concentration ranges suitable for multiplex plasma protein quantitation. Additionally, each SIS PrEST standard can be individually quality assured and quantified by mass spectrometry upfront any spike-in experiment thanks to the use of a common tag sequence. This guarantees that their individual quantities are anchored to the same common point [45]. The quantification is anchored to an ultra-purified tag protein that has been subjected to amino acid analysis to determine its concentration on an absolute scale. Moreover, with the proteotypic peptides being released upon digestion, the quantitative bias between different peptides has previously been shown to be reduced [34]. It can also efficiently account for digestion accuracy biases when using different proteases [53]. This also allows for the unbiased evaluation of the peptides present in the PrEST sequence during method development. When working with high multiplexes of SIL peptides, an unbiased evaluation of peptides can be costly, as there is a need to specifically synthesize the peptides to be interrogated. This can lead to the exclusion of well-performing peptides due to their presumed unsuitability as quantitative proteotypic peptides. This can be avoided when generating assays using SIS PrESTs. However, the peptide selection remains limited to the peptides present in the PrEST sequence, shifting the focus from singular peptides to protein regions. The use of spiked standards at the protein level has also been shown to allow for short digestion protocols (≤60 minutes) with equally high precision as for overnight digestions (12 h) [35, 37]. Additionally, the SIS PREST's ability to deliver robust readouts even after repeated freeze-thaw cycles suggests that this methodology, where samples are spiked with internal standards prior to enzymatic digestion, can help facilitate the introduction of LC-SRM/MS into a clinical setting. It has also been shown that the SIS PrEST can be stored long-term in a vacuum-dried

state for addition-only protocols, which could prove an attractive strategy for clinical applications [37].

In this study, the new methodology was used to determine concentrations of apo(a) as well as the average number of $K4_2$ repeats by quantifying K4-specific peptides in a cohort of 90 individuals without any known disease [46]. In this relatively small cohort with a limited variance in $K4_2$ repeats, it was not possible to detect the negative correlation between the average number of $K4_2$ repeats and the apo(a) concentrations previously observed [10]. Despite not revealing the absolute isoform size of the various populations of circulating apo(a) molecules in blood, this approach for estimating the average circulating isoform size of apo(a) may yet provide useful insights into VTE risk, as described by other studies [54]. As the strategy allows for multiplex analysis of the average isoform size together with quantitative analysis of other proteins, compared to single plex orthogonal methods for determining the isoform size in absolute numbers, it can therefore be deployed in clinical settings without requiring additional experiments to be performed.

The negative correlation could be explained by more efficient hepatic secretion of the shorter isoforms of apo(a) since the production rather than catabolism of Lp(a) is the major determinant of the plasma concentrations. In the relatively small study cohort, visual inspection indicates a negative correlation between apo(a) concentrations and the number of K42 repeats among individuals with more than 13–14 repeats, while a lower number of repeats does not seem to influence the plasma levels. Larger study cohorts are needed to investigate if there is a threshold for the inverse association between the number of repeats and plasma apo (a) concentrations.

The developed assay allows for accurate measurements of molar concentrations of apo(a) and size of apo(a) in an assay suitable for clinical studies. Further studies are needed in larger cohorts with genetic determinations of $K4_2$ copy numbers to allow comparison of apo(a) levels with average apo(a) isoform size and LPA genetics.

## Supporting information

**S1 Table. Sequence of SIS PrEST mapping to apo(a).**
(XLSX)

**S2 Table. LC gradient used for analysis on Agilent 6490.**
(XLSX)

**S3 Table. LC gradient used for analysis of 90 plasma samples.**
(XLSX)

**S4 Table. Transitions monitored for analysis of 90 plasma samples.**
(XLSX)

**S5 Table. Prototypic peptides from the plasminogen domain of apolipoprotein(a).**
(XLSX)

**S1 Data. Reverse standard curves.**
(PDF)

**S1 Graphical abstract.**
(TIF)

## Acknowledgments

We acknowledge the entire staff of the Human Protein Atlas Protein Factory at Albanova.

## Author Contributions

**Conceptualization:** Andreas Hober, Björn Forsström, Jan Oscarsson, Fredrik Edfors, Tasso Miliotis.

**Data curation:** Andreas Hober, Björn Forsström.

**Formal analysis:** Andreas Hober, Mirela Rekanovic, Sara Hansson, David Kotol, Fredrik Edfors, Tasso Miliotis.

**Investigation:** Andreas Hober, Fredrik Edfors.

**Methodology:** Andreas Hober, Björn Forsström, Andrew J. Percy, Mathias Uhlén, Jan Oscarsson, Fredrik Edfors, Tasso Miliotis.

**Project administration:** Tasso Miliotis.

**Resources:** Björn Forsström, Mathias Uhlén, Fredrik Edfors, Tasso Miliotis.

**Supervision:** Andreas Hober, Björn Forsström, Sara Hansson, Mathias Uhlén, Fredrik Edfors, Tasso Miliotis.

**Validation:** Andreas Hober.

**Visualization:** Andreas Hober.

**Writing – original draft:** Andreas Hober, Jan Oscarsson, Fredrik Edfors, Tasso Miliotis.

**Writing – review & editing:** Andreas Hober, Mirela Rekanovic, Björn Forsström, Sara Hansson, David Kotol, Andrew J. Percy, Mathias Uhlén, Jan Oscarsson, Fredrik Edfors, Tasso Miliotis.

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
