## [Decision Letter · Decision Letter 0]

18 Jul 2022

PONE-D-22-14035Targeted Proteomics Using Stable Isotope Labeled Protein Fragments Enables Precise and Robust Determination of Total Apolipoprotein(a) in Human PlasmaPLOS ONE

Dear Dr. Miliotis,

Thank you for submitting your manuscript to PLOS ONE. After extensive review, we feel that it has merit but does not fully meet PLOS ONE’s publication criteria as it currently stands. As you can see, there are a variety of comments and concerns provided by the reviewers below. Therefore, we invite you to submit a revised version of the manuscript that addresses the points raised during the review process.

We look forward to receiving your revised manuscript.

Kind regards,

Jon M. Jacobs, Ph.D.

Academic Editor

PLOS ONE

Journal Requirements:

"I have read the journal's policy and the authors of this manuscript have the following competing interests: M.U. is a co-founder of Atlas Antibodies AB."

Reviewers' comments:

Reviewer's Responses to Questions

**Comments to the Author**

1. Is the manuscript technically sound, and do the data support the conclusions?

Reviewer #1: Partly

Reviewer #2: Yes

Reviewer #3: Partly

2. Has the statistical analysis been performed appropriately and rigorously? 

Reviewer #1: Yes

Reviewer #2: Yes

Reviewer #3: Yes

3. Have the authors made all data underlying the findings in their manuscript fully available?

Reviewer #1: Yes

Reviewer #2: Yes

Reviewer #3: Yes

4. Is the manuscript presented in an intelligible fashion and written in standard English?

Reviewer #1: Yes

Reviewer #2: Yes

Reviewer #3: Yes

5. Review Comments to the Author

Reviewer #1: Summary:

The authors present a novel approach to quantify apolipoprotein(a) by LC-SRM/MS as a surrogate measure of Lp(a) in plasma using SIS-PREST standards spiked in the samples. The method involves spiking of the samples with SIS-PREST standards, digestion of the proteins, clean-up, and concentration by SPE and injection on LC-MS/MS using a targeted method to measure peptides from 1) the repeatable KIV type 2 domain to determine the average apo(a) isoform size and 2) from the singular Protease Domain to measure the absolute concentration of apo(a). The authors present the development and validation of the method as well as a comparison with an ELISA sandwich assay to measure Lp(a) in a study of 100 individual plasma samples.

The research is timely, and the article is well written, the approach is interesting and perspectives for multiplex analyses are highly relevant to the current context of Lp(a) measurements.

Major comments:

Although the validation of the LC-MS/MS method seems to have been performed thoroughly and according to relevant standards for LC-MS/MS method validation, I think some validation points are still missing from the article in its current state.

In particular, there is no assessment of the LC-MS/MS method accuracy. I assume the authors intended to validate accuracy of their assay using the comparison to an ELISA assay but there are some issues with this approach in my opinion. First, the ELISA kit used for the comparison does not provide results in nmol/L while this is the current recommendation for Lp(a) measurements in clinical routine as stated in the introduction by the authors. Moreover, it is unclear if this assay is traceable to the current reference for Lp(a) and if its accuracy has been validated. Secondly, because the authors used an ELISA comparison assay that does not provide results in nmol/L but in Unit/L, they value assigned these calibrators using their own LC-MS/MS method and the SIS-PREST strategy they developed. I think this is a major issue because they are calibrating both their LC-MS/MS method and the comparison method using the exact same calibration system. As a consequence, the comparison of the results from the two methods only shows how good their calibration system is, not how accurate the LC-MS/MS method is. Therefore, these results are not showing method accuracy. To circumvent this issue, I would recommend using an ELISA assay that provides results in nmol/L and is traceable to the WHO/IFCC SRM-2B reference to perform this comparison again. I would also recommend using the Deming regression model with associated error to present the comparison data rather than simple plot as in current article. Finally, the LC-MS/MS method includes a SPE step that is known to have potential recovery issues and the authors did not present data on the assessment of digestion completeness and parallelism to validate the method. I think spiking experiments to validate recovery across the whole LC-MS/MS digestion procedure are needed to finalize the validation in addition to the comparison with the ELISA assay in order to assess the LC-MS/MS method accuracy.

In addition, I think the authors need to clarify and homogenize their statement on the measurement of apo(a) isoform size by LC-MS/MS because it is inconsistent across the article. Indeed, it is of major importance to highlight that measuring a peptide on the KIV2 domain of Lp(a) by LC-MS/MS only provides a measure of the average apo(a) isoform size because 80% of individuals are heterozygous for the LPA gene. It is therefore relevant for only the estimated 20% homozygote individuals. The authors report a cluster of values that behave differently, I think it could be interesting to investigate whether these individuals are homozygote or heterozygote by measuring the apo(a) isoform size using gel electrophoresis as originally reported by Marcovina and colleagues. Moreover, the relevance of the average apo(a) isoform size parameter is still quite debated. Here, the authors present a study on 100 individual plasma samples and investigate the correlation between apo(a) isoform size (which should be labelled average isoform size to clarify) and apo(a) concentration. I think this study is interesting but discussion on the results obtained does not mention the issue of the average isoform size and only identifies the small sample size as a limitation to results interpretation. It also only briefly discusses results obtained by other studies who already investigated this comparison. I think the article would greatly benefit from a more thorough discussion on the aforementioned point.

Finally, I think that overall, the discussion is missing substance and would greatly benefit from some additional points. There are now several methods published on the measurement of Lp(a) by LC-MS/MS and the authors did not discuss the benefits or limitations of their method compared to the others already published. The choice of the peptide is also singular compared to the other methods and a more detailed discussion on the choice of this peptide rather than the LFLE peptide most method used would be beneficial. The benefits of the SIS-PREST approach compared to classic SIL peptides could be of interest too.

Minor comments:

1. Some more explanations as to the way SIS-PREST are digested and used would be beneficial in the introduction.

2. The way the SIS-PREST calibration standards are prepared is unclear. How many calibration points are there? Are the standards digested in duplicate or triplicate?

3. How was the SIS-PREST standard concentration determined? I think it would be beneficial to detail this information more

4. For the intermediate precision assessment, it seems like there is a drift in the values after 5 days. Did the authors perform tests to assess if the slope is significant? Were the calibration standards freshly prepared for all assays of the precision study? It looks like the standards used for all five assays were the same and that there may be a stability issue.

5. I am quite unclear as to why there are 2 different LC-MS/MS injection methods reported for the reference samples and healthy samples. I think the distinction between these samples and their role in the study is missing in the article.

6. Figure 1.A) legend mentions 3 different SIS-PREST peptides but only two are showing on the figure. Figure 1.B) Please also clarify “average” number of K4 repeats.

7. Figure 2: I think including the 2SD interval on the figures would be beneficial.

8. Figure 3: I think showing the identity line and using a Deming regression model would be better to show the comparison data. Also, it is not explicitly explained why the concentrations were log transformed, please clarify.

9. Figure 4B and 4C are not very clear and would benefit from some improvements on readability.

10. Table 2: How was de prediction error determined?

11. Graphical abstract: mention “average” number of K4 repeats

12. Supplementary Data 1: What is the part on reproducibility about? Was this assessed?

13. Supplementary Table: Could these be all merged into a single file for simplicity and more practicality?

Reviewer #2: This manuscript by Hober et al. focusses on the use of mass spectrometry to measure Lp(a) concentrations and apo(a) size polymorphism in human plasma samples. The work is conducted adequately BUT identical studies using the same methodologies have been developed and published in recent years. These studies are not even mentioned in the manuscript except for that of Lassman (2014). The only noticeable difference between this new study and the published ones is that the authors used a distinct proteotypic peptide for apo(a) concentration determination.

Lassman (2014) is cited but only to mention that counting the KIV domains by MS is do-able. It is indeed do-able since it has been done in the past and has been used by many groups for many different studies.

Please refer for instance to the most recent reports (among others):

PMID33517366 (Marcovina 2021)

PMID34314498 (Blanchard 2021)

PMID 32404332 (Blanchard 2020)

Reviewer #3: This is a well-written manuscript from a strong and experienced team of which some co-authors I know by relevant publications in proteomic field and HUPO consortium. The authors offered an unconventional approach for the absolute quantification of plasma circulating supramolecular complex, which is LDL-like particle, or Lp(a). The proposed approach permits simultaneous measurement of concentration Apolipoprotein A and estimation of constituent KIV2 repeats, the number of which may vary broadly between subjects. The established method has been validated by traditional ELISA and demonstrated satisfied correlation and overlap within relevant linear range of concentrations.

1. The number of calibrating levels for the developed LC-SRM (page 6, Material and Methods, section “Standard curve”) is not indicated. It is essential to indicate the total number of calibrating levels and the whole range expected to cover by calibration, albeit readers can find this information in Supplementary data. Such information should be defined in the main text first in order to understand calibration range.

2. Why retention time window is so wide (5 minutes) for the developed LC-SRM (Supplementary Table 4)? The wider retention time window, the more concurrent transitions are collected within certain window and less dwell time you obtain per a transition. For example, there are transitions with only 0.866 ms dwell time. Is it enough for the efficient signal accumulation? The authors used quite robust and reproducible UPLC system giving, probably, sharp peaks with 6-10 s full width (I assume) so they could define a narrow window within 1 minute or less.

3. Quality control of the obtained signal included curation of the “peak shape” as defined in the “Data analysis” section (page 10). It is absolutely unclear and insufficient information of what does the ‘peak shape’ mean? Chromatographic peak consist of many different but well-defined features and the ‘peak shape’ comprises of peak width, asymmetry, tailing, etc. I suggest authors have to provide some more details about the ‘peak shape’ criteria they used for the quality control.

4. Obviously, authors employed the Skyline tool for the selection of proper peptides and transitions. In this respect, authors have to indicate such criteria of selection (min-max length, charge states selected (because they used peptide-quantifier with the charge z = 3+), fragment ions type (b, y, a (included?)) and m/z range, definition and monoisotopic or average masses (because they utilized low-resolution instrument), etc. It is essential to apply such criteria in proper way.

5. No doubts as to perfectly done research and development, I suggest the lack of some important information. The authors used two different PrEST as illustrated in Figure 1 (page 11) and stated in the “Results” section (targeted by using two separate SIS recombinant proteins…) which is caused by the interest in specific calculation of K4 domain copy numbers per a molecule and by estimation of the total Apolipoprotein A concentration. However, I found only one of the PrEST constructions in the Supplementary Table 1, which corresponds to 1870-2006 region of canonical Apo A sequence and embodies peptide EAQLL… for particular protein quantification. But where is the second PrEST construction with the embedded K4 domain-specific sequence (GTYS…) for calculation of KIV2 copies?

6. I accept the defined LOD and LOQ calculated by authors and proposed approach for their calculation. I do not usher to refine and revise them, but I think LOD and LOQ are incorrect and inaccurately calculated. Based on my own and our laboratory experience, we regularly apply recommendations of IUPAC and approved guideline for Evaluation of Detection Capability for Clinical Laboratory Measurement. According such recommendation, you have to establish LOB (limit of blank) first which permits you establishing of LLOD (lowest limit of detection), then LOD and, finally, LOQ. Moreover, you need ‘k’ and ‘b’ coefficient of your linear regression to calculate LOQ correctly: LOQ=LOD+10σc(|b|⁄k)/ Eventually, double LOQ gives Method Limit (ML) value.

7. The selected peptide EAQLLVIENEVCNHYK bears N-terminal glutamic acid which intends to cyclization favorable at already pH=4; internal glutamine and two asparagine residues, which are readily deamidated even at mild conditions; and internal cysteine residue which is modified during alkylation and assumes complete reaction (alkylation) recovery else fraught with inaccurate quantification. It seems that the peptide is hard to control. Why did the authors choose such ‘uncomfortable’ peptide as the main source for quantification? Why, for example, peptide VILGAHQEVNLESHVQEIEVSR was ignored?

8. The lowest and the highest measured concentrations in assay of ELISA vice the developed LC-SRM seem like outliers (Table 3). If these two points (samples S17 and S20) are eliminated, the corrected R would be more than 0.99. Is it because these measurements are close to limits of detection (upper and lowest) and linearity of LC-SRM and because lower robustness of ELISA (for example, I know that immunoturbidimetry used for Apo A assay is of poor robustness and low accuracy)? Does it mean that the developed LC-SRM is less applicable in clinics compare to ELISA? Does such broad difference (more than 50% in some cases) can be managed in corrective manner to obtain serial results matching stronger between two methods (immunoassay and SRM)?

9. A little more information is essential for samples collected and used for method development, validation and random measurements because Apo A concentration is highly sensitive to fasting before samples collection, smoking before collection, BMI, age and (optional) comorbidities. I do nor require clinical records and history of subjects under consideration, but at least minimal information (as has been touched above) is obligatory to understand possible reason of enormous difference between samples and method utilized (since ELISA is obviously sensitive to deep freezing and thawing because proteins undergo increased denaturation and loss their three-dimensional stability during freezing at low temperatures, which is critically important and may significantly affect the ELISA results. That is why we can occasionally inspect enormous difference between two methods for some instances).

Overall, I liked this paper while reading. It was interesting despite the research seems simple. Results of KIV2 copies calculation are intriguing and discovered lack of correlation with the total concentration of Apo A is fascinating. My last question is only of my interest and does not require and modifications in the main text of the paper: why authors went through the synthesis of recombinant proteins fragment (PrEST) instead of preliminary selection proper peptides and their synthesis with terminal isotope-labeled residues? What is the main advantage of PrEST compared to conventional bioinformatic selection and routine synthesis, which are cheaper and easier to follow?

6. PLOS authors have the option to publish the peer review history of their article (what does this mean?). If published, this will include your full peer review and any attached files.

Reviewer #1: No

Reviewer #2: No

Reviewer #3: **Yes: **Arthur T. Kopylov

---

## [Author Response · Author response to Decision Letter 0]

16 Sep 2022

Response to reviewer comments has been uploaded as a separate file ("Response to reviewers").

---

## [Decision Letter · Decision Letter 1]

18 Oct 2022

PONE-D-22-14035R1Targeted Proteomics Using Stable Isotope Labeled Protein Fragments Enables Precise and Robust Determination of Total Apolipoprotein(a) in Human PlasmaPLOS ONE

Dear Dr. Miliotis,

Thank you for submitting your manuscript to PLOS ONE. After consideration again by the reviewers, there still remains additional concerns prior to acceptance of the manuscript.  See comments below.  Please submit your revised manuscript by Dec 02 2022 11:59PM. If you will need more time than this to complete your revisions, please reply to this message or contact the journal office at plosone@plos.org. Please include the following items when submitting your revised manuscript:A rebuttal letter that responds to each point raised by the academic editor and reviewer(s). You should upload this letter as a separate file labeled 'Response to Reviewers'.A marked-up copy of your manuscript that highlights changes made to the original version. You should upload this as a separate file labeled 'Revised Manuscript with Track Changes'.An unmarked version of your revised paper without tracked changes. You should upload this as a separate file labeled 'Manuscript'.

We look forward to receiving your revised manuscript.

Kind regards,

Jon M. Jacobs, Ph.D.

Academic Editor

PLOS ONE

Reviewers' comments:

Reviewer's Responses to Questions

**Comments to the Author**

1. If the authors have adequately addressed your comments raised in a previous round of review and you feel that this manuscript is now acceptable for publication, you may indicate that here to bypass the “Comments to the Author” section, enter your conflict of interest statement in the “Confidential to Editor” section, and submit your "Accept" recommendation.

Reviewer #1: (No Response)

Reviewer #2: All comments have been addressed

Reviewer #3: All comments have been addressed

2. Is the manuscript technically sound, and do the data support the conclusions?

Reviewer #1: Partly

Reviewer #2: Yes

Reviewer #3: Yes

3. Has the statistical analysis been performed appropriately and rigorously? 

Reviewer #1: Yes

Reviewer #2: (No Response)

Reviewer #3: Yes

4. Have the authors made all data underlying the findings in their manuscript fully available?

Reviewer #1: Yes

Reviewer #2: Yes

Reviewer #3: (No Response)

5. Is the manuscript presented in an intelligible fashion and written in standard English?

Reviewer #1: Yes

Reviewer #2: Yes

Reviewer #3: Yes

6. Review Comments to the Author

Reviewer #1: Although the authors addressed some of my previous comments and suggestions, I do not think the quality of the article has sufficiently improved yet to be released for publication. Some of my main concerns have not been addressed and I think the current article is still missing some important discussion points:

1. I understand the authors response to my comment on the choice of the comparison ELISA assay for the validation of their LC-MS/MS method although I do not agree. Assessing accuracy means assessing closeness of agreement of the value measured with the new method versus the true value of the analyte in the sample. The true value cannot be measured and only estimated within an error range. Metrology institutes and international instances for method accuracy work worldwide on developing standards, measured with highly precise and highly accurate methods, traceable to the international system of units to determine a value as close as possible to the true value for an analyte so that secondary methods like ELISA can use them as calibrators and be ensured accuracy. The SRM-2B is a very imperfect matrix-based material, a pool of plasma indeed, but it has one advantage: it has an assigned reference value, internationally agreed on and used as a general anchor for most Lp(a) assays worldwide. Using it as a comparison point, although it will not be accuracy in its strictest sense since SRM-2B is not traceable to the SI, is still the best comparison point available for Lp(a).

My suggestion and comments were not this though. SRM-2B is a precious material not easily accessible for studies and could understandably not be used for this research. However, multiple commercial assays for ELISA are directly traceable to SRM-2B, and provide results directly in nmol/L, which is the best solution. The fact that the authors used a benchmark assay for their comparison is not my point. I fully trust the good sense of the researchers to pick an assay performing well. My point was that they should chose an assay traceable to the SRM-2B, the only available international reference for Lp(a), so that they compare their LC-MS/MS method to an ELISA which accuracy has been proven and certified. Comparing to an ELISA without stating its traceability does not provide any information on accuracy, it only provides information that the LC-MS/MS is comparable to this specific ELISA.

Based on the authors response, I seem to understand that the Mercodia ELISA assay they used was verified in some ways against commercial controls. The traceability of this ELISA kit to the SRM-2B, if available, should be mentioned clearly in the main text. If the ELISA used is not traceable to the SRM-2B, this issue should be thoroughly discussed in the main text and the associated limitation regarding claims for accuracy clearly stated. Similarly, the controls used to check this test and their traceability should be mentioned in the main text.

I strongly insist on addressing this concern within the main text. Authors mention they updated this limitation in the main text, I do not see these changes. I think they should mention their benchmark so that they can defend within the main text and to all readers the choice of this ELISA and the reason why they are confident in their assessment of accuracy by comparing their LC-MS/MS to this ELISA specifically.

2. I disagree with the authors’ response regarding the measure of average apo(a) isoform size. I did understand the goal of measuring the total number of KIV-2 circulating in blood and measuring this is, indeed, interesting in some ways. However, what is important about apo(a) isoform size is that it will impact Lp(a) particle size, which will have consequences on its metabolism. What is relevant is therefore the size of the particles circulating, of which apo(a) isoform is only a surrogate measure. Knowing the total average amount of KIV-2 does not provide this information.

The reason I insist is precisely that people tend to get confused on this point, which I think us as researchers are responsible for making clear. I do not question the quality of the method to measure the average apo(a) isoform size but ask that this nuance in terms of clinical usefulness be mentioned in the main text clearly.

Moreover, there do exists a method to assess the distribution of apo(a) isoform size within a sample: gel electrophoresis as mentioned previously by the authors. I was not requesting that the authors run this method on their sample, and I understand it is not feasible for these study samples given small amounts available. However, mentioning the limitation of average apo(a) isoform size vs actual apo(a) isoform size distribution measurement and stating these limitations in the discussion is feasible. There is ample available bibliography on this subject.

3. The authors state in their response that they expanded the discussion to highlight the limitations of this study. I do not see the difference in the revised manuscript. I maintain my comment that the discussion in its current form still lacks substance.

First of all, as mentioned by reviewer 2, the authors do not cite nor discuss work previously published on LC-MS/MS methods to measure apo(a). The mention of previously published methods only appears in the introduction and is quite short and feels very expeditive. I think expending on what the 3 other groups with existing methods have already done, at least in a few sentences, would be a good idea for the introduction. Even though this method is definitely interesting with SIS-PrEST technology and its value-added for quantification, the way the introduction is written sounds like this is the first ever LC-MS/MS method on Lp(a), which it is not.

Overall, my point is: ok the SIS-PrEST approach is an interesting approach, but what is new for Lp(a)? In the end, what is the point of using this method versus another multiplex one developed by Blanchard and colleagues for example who use cheap and easy access SIS peptides, have good accuracy and quick method running in clinical setting already? Why go through the hurdle of using SIS-PrEST? How expensive is the use of SIS-PREST? You mention its potential use in clinical setting, would that be financially robust and feasible? What are the performances of this specific method compared to that of Marcovina and colleagues in terms of reproducibility, turn-around time, precision? What about discussion on the choice of this long peptide, potentially unstable as very interestingly pointed out by reviewer 3? What about digestion completeness and accuracy bias due to different steps? The authors answered to me that SIS-PrEST takes this into account, I think this answer should typically be included in the discussion as it is relevant to the readers too. What about the limitations of this method in terms of practicality in clinical laboratories? I mentioned prices of the SIS-PrEST, what about ease of access? Batch to batch reproducibility and purity? Stability over long period of storage?

In my opinion, all these points should be extensively mentioned and discussed in the article to improve its quality, interest, and general impact to the scientific community.

Minor comments:

1. Page 14 – Figure 3: Where is the discussion regarding the fact that the error range is quite large and that there is significant deviation from unity at low concentrations?

2. Page 16 – Line 2 (discussion): Average isoform size.

Reviewer #2: The authors have satisfactorily answered to my initial comments. I than them for their response and hope that their article will meet its readership.

Reviewer #3: The authors satisfied all questions addressed. I have also reviewed data submitted to Panorama (as declared in the ‘Data Availability” statement) and I was satisfied by the presented results concerning signal stability and robustness. In the revised paper the Authors also unfurled the advantage of SIS PrEST approach as I asked them before, and added relevant references. I hope, the Authors would update the review about this technique and achievements in someone future specific paper.

The Authors also made essential amendments in the Supplementary to display accurately the design of SIS PrEST experiment as I claimed before. I also marked more details regarding the comparison of efficiency between ELISA and SRM approaches and comprehensive calculation of the true concentration of calibrants. I have no doubts and specific question regarding the revised version of manuscript. Only minor corrections and amendments are needed. Specifically:

1. ‘SI-units’ abbreviation appeared the first time in page 13 and should be defined;

2. ‘…protein-level spike-ins of standards rather…’: spike-ins’ is probably mistyped and should be ‘spike-in’;

3. ‘Regionala etikprövnignsnämnden’ on page 6: the ‘Etikprövnignsnämnden’ should begin from the capital letter “E” as on page 5;

4. I would suggest to place a part the ‘Results’ section (Peptide ions with a length of 5-25 amino acids and a precursor charge state z = 2,3 and product ion charge state z = 1,2 were screened. Only b- and y-ions were included in the analysis. (33, 34)) in the ‘Material and Methods’ section as a better place.

5. ‘Discussion’ section, page 17: ‘…variance in KIV2 repeats, it was not possible to detect the previously observed negative correlation between number of KIV2 repeats and apo(a) concentrations previously observed (10).’ – I think, one of ‘previously observed’ is redundant.

7. PLOS authors have the option to publish the peer review history of their article (what does this mean?). If published, this will include your full peer review and any attached files.

Reviewer #1: No

Reviewer #2: No

Reviewer #3: **Yes: **Arthur T. Kopylov

---

## [Author Response · Author response to Decision Letter 1]

28 Dec 2022

Response to reviewers comments have been uploaded as a separate file.

---

## [Decision Letter · Decision Letter 2]

1 Feb 2023

Targeted Proteomics Using Stable Isotope Labeled Protein Fragments Enables Precise and Robust Determination of Total Apolipoprotein(a) in Human Plasma

PONE-D-22-14035R2

Dear Dr. Miliotis,

We’re pleased to inform you that your manuscript has been judged scientifically suitable for publication and will be formally accepted for publication once it meets all outstanding technical requirements.

Kind regards,

Jon M. Jacobs, Ph.D.

Academic Editor

PLOS ONE

Additional Editor Comments (optional):

Reviewers' comments:

Reviewer's Responses to Questions

**Comments to the Author**

1. If the authors have adequately addressed your comments raised in a previous round of review and you feel that this manuscript is now acceptable for publication, you may indicate that here to bypass the “Comments to the Author” section, enter your conflict of interest statement in the “Confidential to Editor” section, and submit your "Accept" recommendation.

Reviewer #1: All comments have been addressed

Reviewer #3: All comments have been addressed

2. Is the manuscript technically sound, and do the data support the conclusions?

Reviewer #1: Yes

Reviewer #3: Yes

3. Has the statistical analysis been performed appropriately and rigorously? 

Reviewer #1: Yes

Reviewer #3: Yes

4. Have the authors made all data underlying the findings in their manuscript fully available?

Reviewer #1: Yes

Reviewer #3: Yes

5. Is the manuscript presented in an intelligible fashion and written in standard English?

Reviewer #1: Yes

Reviewer #3: Yes

6. Review Comments to the Author

Reviewer #1: The authors addressed my most important comments. I have only minor comments left:

- Page 4, Line 16: Clarify sentence

- Page 4, Line 20: There is a typo on “proteotypic peptide”

- Page 4, Line20: Remove “Accurate” from “thereby enables accurate measurement of the molar apo(a) levels in plasma.” Accuracy is not related to the choice of the peptide per se.

- Throughout the whole manuscript: Homogenize KIV or K4 abbreviation.

- Page 12, Line 1: There is a typo on “proteotypic peptide”

Reviewer #3: The authors have replied my comments with efforts during two rounds of the hard peer-review. I have no more doubts and more comments. I would agree to publish this paper.

7. PLOS authors have the option to publish the peer review history of their article (what does this mean?). If published, this will include your full peer review and any attached files.

Reviewer #1: No

Reviewer #3: **Yes: **Arthur T. Kopylov

---

## [Editor Report · Acceptance letter]

3 Feb 2023

PONE-D-22-14035R2 

Targeted Proteomics Using Stable Isotope Labeled Protein Fragments Enables Precise and Robust Determination of Total Apolipoprotein(a) in Human Plasma 

Dear Dr. Miliotis:

I'm pleased to inform you that your manuscript has been deemed suitable for publication in PLOS ONE. Congratulations! Your manuscript is now with our production department. 

Kind regards, 

on behalf of

Dr Jon M. Jacobs 

Academic Editor

PLOS ONE